# Galactofuranose (Gal*f*)-containing sugar chain contributes to the hyphal growth, conidiation and virulence of *F. oxysporum* f.sp. *cucumerinum*

Hui Zhou[1], Yueqiang Xu[1,2], Frank Ebel[3], Cheng Jin[1,2,4]*

**1** State Key Laboratory of Mycology, Institute of Microbiology, Chinese Academy of Sciences, Beijing, China, **2** University of Chinese Academy of Sciences, Beijing, China, **3** Institute for Infectious Diseases and Zoonoses, LMU, Munich, Germany, **4** National Engineering Research Center for Non-food Bio-refinery, Guangxi Academy of Sciences, Nanning, China

* jinc@im.ac.cn

**Data Availability Statement:** All relevant data are within the paper and its Supporting Information files.

## Abstract

The ascomycete fungus *Fusarium oxysporum* f.sp. *cucumerinum* causes vascular wilt diseases in cucumber. However, few genes related to morphogenesis and pathogenicity of this fungal pathogen have been functionally characterized. BLASTp searches of the *Aspergillus fumigatus* UgmA and galatofuranosyltransferases (Gal*f*-transferases) sequences in the *F. oxysporum* genome identified two genes encoding putative UDP-galactopyranose mutase (UGM), *ugmA* and *ugmB*, and six genes encoding putative Gal*f*-transferase homologs. In this study, the single and double mutants of the *ugmA*, *ugmB* and *gfsB* were obtained. The roles of UGMs and GfsB were investigated by analyzing the phenotypes of the mutants. Our results showed that deletion of the *ugmA* gene led to a reduced production of galactofuranose-containing sugar chains, reduced growth and impaired conidiation of *F. oxysporum* f. sp. *cucumerinum*. Most importantly, the *ugmA* deletion mutant lost the pathogenicity in cucumber plantlets. Although deletion of the *ugmB* gene did not cause any visible phenotype, deletion of both *ugmA* and *ugmB* genes caused more severe phenotypes as compared with the ΔugmA, suggesting that UgmA and UgmB are redundant and they can both contribute to synthesis of UDP-Gal*f*. Furthermore, the ΔgfsB exhibited an attenuated virulence although no other phenotype was observed. Our results demonstrate that the galactofuranose (Gal*f*) synthesis contributes to the cell wall integrity, germination, hyphal growth, conidiation and virulence in *Fusarium oxysporum* f.sp. *cucumerinum* and an ideal target for the development of new anti-*Fusarium* agents.

## Introduction

Galactofuranose (Gal*f*) is a component of several polysaccharides and glycoconjugates comprising major portions of the cell surface in fungi, bacteria, trypanosomatids and nematodes. In bacteria, Gal*f* constitutes a key part of the mycobacterial cell wall and occurs in

**Funding:** This work was supported by the National Natural Sciences Foundation of China (31630016 and 31320103901) and partially supported by Bagui Scholar Program Fund (2016A24) of Guangxi Zhuang Autonomous Region to CJ. the funders play no role in the study design, data collection and analysis, decision to publish, or preparation of the manuscript.

**Competing interests:** The authors have declared that no competing interests exist.

lipopolysaccharide (LPS) O-antigen domains, extracellular capsules, and polysaccharides [1]. In certain filamentous fungi, Gal*f* is a major component of the cell wall and structural glycoproteins [2–7]. In pathogenic protozoa, Gal*f* residues are key components of surface glycosylphosphoinositol (GPI)-anchored glycoconjugates in *Leishmania* and *Trypanosoma cruzi* [8–10]. In nematode such as *Caenorhabditis elegans* Gal*f* residues are key components of surface coat [11]. In contrast, higher animals including humans and plants have no Gal*f*-containing sugar chains. Therefore, inhibition of the biosynthetic pathway of Gal*f*-containing sugar chains is an attractive target for the development of medicines and agricultural chemicals inhibiting pathogenic fungi, bacteria and parasites without any side effects.

All Gal*f*-containing sugar chains are synthesized using UDP-Gal*f* as a sugar donor. The biosynthesis of UDP-Gal*f* begins with the conversion of UDP-glucose to UDP-Gal*p* by UDP-glucose 4-epimerase (UGE), then UDP-Gal*p* is converted to UDP-Gal*f* by UDP-galactopyranose mutase (UGM), and eventually Gal*f* is transfered from UDP-Gal*f* to Gal*f*–containing sugar chains by galactofuranosyl(Gal*f*)-transferases. In fungi, the site of Gal*f*-containing sugar chains biosynthesis is the internal lumen of the Golgi apparatus. Therefore, a UDP-Gal*f* transporter is necessary to transport the synthesized UDP-Gal*f* from the cytosol to the inner lumen of the Golgi apparatus [12].

The genes responsible for the Gal*f* biosynthesis have been identified in *Aspergillus fumigatus*, *A. nidulans* and *A. niger*, which include a similar gene sets of UDP-glucose 4-epimerases (UgeA), UDP-galactomutases (UgmA), UDP-Gal*f*- transporters (UgtA and UgtB), and Gal*f*-transferases [13–24]. In *A. nidulans*, deletion of the *ugeA*, the gene encoding UDP-glucose 4-epimerase, leads to highly branched hyphae and reduced conidiation [15]. The *A. niger* Δ*ugmA* strain is highly sensitive to the chitin binding agent calcofluor white, suggesting that synthesis of a Gal*f*-containing sugar chain is involved in the biosynthesis of cell wall components [22]. Deletion of the gene encoding UGM (*glfA*), the UDP-Galactofuranose transporter (*glfB*), or the Gal*f*-transferase GfsA in *A. fumigatus*, leads to the absence of the galactofuran side chains of galactomannan and is associated with increased susceptibility to antifungals, attenuated virulence, and reduced growth [16,19,20,24]. More recently, three putative Gal*f*-transferases, GfsA, GfsB and GfsC, were identified in *A. niger*. Analysis of the single, double and triple mutants revealed that GfsA together with GfsC are most important for the biosynthesis of Gal*f*-containing sugars and multiple *gfs* gene deletions caused a severe cell wall stress response [24]. These reports reveal that a lack of Gal*f*-containing sugar chains affects normal hyphal growth, cell wall structure and conidiation in filamentous fungi.

*Fusarium oxysporum* is a relevant plant pathogen causing *Fusarium* wilt syndrome in more than 120 different hosts and have been ranked in the top 10 plant fungal pathogens [25]. Although β1-6-linked Gal*f* residues have been identified in the cell wall of *Fusarium* [26], the significance of Gal*f* sugar residues in *F. oxysporum* remained to be unknown. Therefore, it is of important to evaluate the significance of Gal*f* biosynthesis in growth and pathogenicity of this species.

In this study, BLASTp searches identified two putative UGM genes, *ugmA* (EGU88002.1) and *ugmB* (EGU83790), and seven putative Gal*f*-transferase genes, EGU86692, EGU81169, EGU76406, EGU88219, EGU80311 and EGU82964 in the genome of *F. oxysporum*. The single and double mutant strains of the UGM genes, as well as the single mutant of the *gfsB* (EGU86692) were obtained in *F. oxysporum* f.sp. *cucumerinum*, a strain infecting cucumber and resulting in serious vascular disease worldwide [27]. By analyzing these mutants, the functions of Gal*f*-containing sugars in growth, conidiation and pathogenicity of *F. oxysporum* f.sp. *cucumerinum* were examined.

## Materials and methods

### Strains, plasmids, and culture conditions

*F. oxysporum* f.sp. *cucumerinum* (CGMCC3.2830, China General Microbiological Culture Collection Center) was used as the wild-type strain (WT) in this work. *F. oxysporum* f.sp. *cucumerinum* strains were cultured at 28˚C in potato dextrose agar (PDA) or in potato dextrose broth (PDB) with shaking at 200 rpm for production of fungal spores and mycelia. Conidia were washed with ddH$_2$O from 7-days culture on plates and counted in a haemocytometer. Mycelia were harvested by filtering the 2-days cultures with two layers of slow qualitative filter paper. The pSCN44 vector harboring a hygromycin gene was used for mutant construction and the pKN vector carrying a neomycin gene was used for generating the double-mutant and the complemented strains. All vectors and plasmids were propagated in *Escherichia coli* DH5α.

### Transformation, selection and characterization of stable transformants

The protoplasts were prepared according to Sarrocco et al. [28] with some modification. One milliliter of $10^9$ ml$^{-1}$ freshly harvested conidia of *F. oxysporum* f.sp. *cucumerinum* was inoculated in 100 ml of YPD medium and incubated at 28˚C, 200 rpm for 5–7 hours. The germinated conidia were collected by centrifugation at 3,000 rpm and washed three times with 0.7 M NaCl, then resuspended by gentle vortexing in an enzyme mixture containing lysing enzyme (10 mg ml$^{-1}$), cellulase (15 mg ml$^{-1}$), and driselase (10 mg ml$^{-1}$) (Sigma-Aldrich) in 0.7 M NaCl and incubated at 30˚C, 100 rpm for about 1 hour. The protoplasts were counted by a hemocytometer and resuspended in STC buffer (1.2 M sorbitol, 10 mM Tris-HCl, pH8, 50 mM CaCl$_2$) in a final concentration of $10^8$ CFU ml$^{-1}$. For the transformation, an aliquot of 200 µl protoplast suspension was gently mixed with 5 µg DNA and 50 µl of 30% PEG solution (30% PEG 8000, 10 mM Tris-HCl, pH8, 50 mM CaCl$_2$) and incubated at room temperature for 20 minutes, then 2 ml of 30% PEG solution were added, mixed carefully and incubated at room temperature for 5 minutes. Finally, 4 ml of STC buffer were added and mixed thoroughly. The transformants were screened using 50 µg ml$^{-1}$ hygromycin B (Sigma-Aldrich) and screened again with 150 µg ml$^{-1}$ hygromycin B. To obtain the double-deletion mutant, G418 was used to screen the mutant strains. For the complemented strains of the double mutants, 100 µg/ml phleomycin was used as the selection marker.

### Targeted gene replacement and complementation

For the Δ*ugmA* and Δ*ugmB* mutant, the 1500-bp fragments of the upstream and downstream flanking sequences of the *ugm* genes were obtained by PCR amplification from wild-type genomic DNA using the primer pairs of ugmu1/ugmd1 and ugmu1/ugmd2 (S1 Table), respectively, and the hygromycin resistance gene (*hyg*) was amplified from the pCSN44 plasmid with the primer pair Hyg1/Hyg2. Then, the inserted cassette was generated by fusing the upstream and downstream flanking regions to the two ends of the *hyg* gene by overlapping-PCR. Finally, the cassette fragments were transformed into the protoplast of the wild-type strain. A double mutant was obtained by deletion of *ugmA* in the Δ*ugmB* mutant. Complementation of the Δ*ugmA* mutant was achieved by introducing pKN-*ugmA*, which was obtained by DNA amplification using primer pair ugmA C-F/ugmA C-R (S1 Table). For the completement of the double mutant, 100 µg/ml of phleomycin was used to screen the transformants. In all cases, HygR, G418R, or PhleoR resistant transformants were obtained and the homologous recombination or complementation events were confirmed by PCR and Southern blotting. The primers and probes used in this study are listed in S1 Table. KOD FX DNA polymerase (TOYOBO) was used in all PCR experiments. Plasmid DNA was extracted using the Plasmid Miniprep Kit

(Axygen, China). Fungal genomic DNA was extracted using the method described previously [29]. Using the same method, the Δ*gfsB* mutant and its complemented strain was constructed using the primers shown in S1 Table. All mutants were confirmed by Southern Blotting.

## Phenotypic analysis of the mutants

A droplet of $1 \times 10^5$ freshly harvested conidia was inoculated at the centre of a plate and cultured at 28˚C. The growth of the colonies was evaluated after 7 days. The conidia production of the strains were quantified in static cultures grown in continuous dark as described previously [30].

To test the sensitivities of the mutants to anti-fungal reagents, a serial dilution of conidia of $1 \times 10^5 - 1 \times 10^2$ were grown at 28˚C on PDA or MM (1 g/L $KH_2PO_4$, 2 g/L $NaNO_3$, 0.5 g/L KCl, 0.5 g/L $MgSO_4$ heptahydrate, 30 g/L sucrose) [31] agar plates supplemented with 100 μg/ml Congo red, 50 μg/ml calcfluor white or 1.2 M sorbitol.

Germination analysis was carried out by inoculating $1 \times 10^5$/ml or $1 \times 10^6$/ml of freshly harvested conidia onto a glass slide with double concave in 50 μl PDB liquid medium. After incubation at 28˚C for specific times, the slides were taken out and observed [32].

## Expression of the *ugm* genes in the WT and mutant strains

The freshly harvested conidia were cultured in PDB medium for 6, 12, 24, 36 and 48 hours. The hyphae were harvested by filtration through Whatman filter paper, washed with sterile water, and then ground under liquid nitrogen using a mortar and pestle. Total RNAs were isolated from ground conidia and mycelia using Trizol reagent (Invitrogen) according to the manufacturer's protocol. Complementary DNA synthesis was generated with 2 μg of RNA using the RevertAid First Strand cDNA Synthesis Kit (ThermoScientific). The primers (ugmA5'/ugmA3' and ugmB5'/ugmB3') used are listed in S1 Table. Whenever possible, the primers spanned an intron. Gene expression was analyzed with 300 nM primers in 20 μl reaction buffer by using EvaGreen qPCR Mastermix (ABM Company) and an iCycler apparatus (Bio-Rad). Cycle conditions were 10 min at 90˚C and 40 cycles of 15 s at 95˚C and 1 min at 60˚C. Single PCR products were confirmed with the heat dissociation protocol at the end of the PCR cycles. For each primer pair, the amplification efficiency was determined by serial dilution experiments, and the resulting efficiency coefficient was used for the quantification of the products. All reactions were performed in triplicate. The relative mRNA expressions were calculated according to $2^{-\Delta\Delta Ct}$ with the transcription elongation factor gene (EF) as the reference.

## Detection of Gal*f* with antibody

The anti-Gal*f* monoclonal antibody AB135-8, which was identified and cloned from a pool of hybridoma cells obtained after immunization of Balb/c mice with killed *A. fumigatus* germ tubes [33]. Resting conidia were incubated in PDB liquid medium for 8 h and fixed with 2.5% p-formaldehyde (PFA) overnight at 4˚C. After fixation, cells were washed with 0.2 M glycine in PBS for 5 min, then sealed with 5% BSA in PBS for 1 h. Cells were incubated with the hybridoma culture supernatant containing AB135-8 at 1:10 dilution in 5% BSA in PBS for 1 h at room temperature. After washing with 1% BSA in PBS, cells were incubated with a goat Dylight 549-conjugated antibody directed against mouse IgG (H+L) diluted 1:2000 in BSA/ PBS. After washing with PBS, cells were visualized with an inverted fluorescence light microscope.

## Virulence

The cucumber seeds (Chinese cultivar "Zhong Nong 16") were processed for accelerating germination as described by Li et al. [34], and the germinated seeds were grown in plastic pots filled with sterilized soil composed of vermiculite and turf. The pots were placed in the greenhouse at approximately 28˚C for a 16 h photoperiod until the two-true leaves stage. For virulence test of Δ*ugmA*, Δ*ugmB* and double mutant, the seedling roots were slightly wounded and the cucumber plants were cultured in 1 g/L Murashigeand Skoog medium (MS) with vitamin supplements containing $2×10^5$ freshly harvested conidia. The pots were placed in green house with the same condition as described above. Twelve plants were used for each treatment. For virulence test of Δ*gfsB*, the slightly wounded seedling roots were immersed in conidia suspension of $5×10^6$/ml for 30 minutes. Subsequently, the infected seedlings were transplanted to plastic pots and placed in green house. Disease severity index (DSI) was calculated as previously described [35] from 2 weeks of inoculation up to 45 days. The plant infection experiments were repeated for three times. DSI was calculated as: DSI = [Σ (Class×Number of plants in that class)/(4×Total number of assessed plants)] ×100.

To examine the invasion of *F. oxysporum* f.sp. *cucumerinum* in plant vascular bundles, the root cross sections of infected plants were observed with a M205A stereoscopic microscope (Leica Microsystems).

To quantify the relative mycelial biomass of the fungal strains in cucumber roots, DNA samples were extracted from the cucumber roots cultured for 30 days after infection and subjected to quantitative PCR (qPCR) using primers EF1a-1/-2 for the transcription elongation factor gene of *F. oxysporum* f.sp. *cucumerinum*. The amounts of DNA templates were normalized using the cucumber actin 1 gene (primers: ACT1F/ACT1R). Three simultaneous replicated amplifications were carried out for each DNA sample, using 20-μL aliquots from a 60-μL mixture, and Bright Green qPCR Mastermix kit (ABM Company). qPCRs were performed in an iCycler apparatus (Bio-Rad) using the following cycling protocol: an initial step of denaturation at 95˚C for 10 min, followed by 40 cycles of 15 s at 95˚C, 60 s at 60˚C. After amplification, a melting curve program was run for which measurements were made for 5 min within a range of 65–95˚C. The DNA concentration of each sample was extrapolated from standard curves, which were developed by plotting the logarithm of known concentrations (10-fold dilution series from 100 ng to 1 ng/20 μL reaction) of *F. oxysporum* f.sp. *cucumerinum* genomic DNA against the Ct values. The relative DNA amounts of the indicated strains were calculated by comparative ΔCt from the mean of two different determinations of the threshold cycle. Each PCR reaction was followed by a melting curve, to assure that there was only one product amplified. All primers used in this assy are listed in S1 Table.

## Results

### Construction of the single and double mutants

BLASTp searches of the *A. fumigatus* UDP-galactopyranose mutase GlfA/Ugm1 (AFU_3G12690) and Gal*f*-transferases in the genome of *F. oxysporum* 5176 identified two putative UDP-galactopyranose mutase genes, *ugmA* (EGU88002.1), *ugmB* (EGU83790) and six putative Gal*f*-transferase genes, *gfsB* (EGU86692), *gfsC* (EGU81169), *gfsE* (EGU80311), *gfsF* (EGU88219), *gfsG* (EGU76406) and *gfsH* (EGU82964) (Table 1). The *ugmA* encodes a predicted protein (519 amino acids) sharing 80% of identity with *A. fumigatus* GlfA, while the *ugmB* encodes a protein (507 amino acids) sharing 71% of identity with *A. fumigatus* GlfA. RT-PCR analysis revealed that both *ugmA* and *ugmB* were expressed in *F. oxysporum* f. sp. *cucumerinum* (S1 Fig). As only GfsB and GfsC, which are belong to GT31-A family together

**Table 1.  Identification of Gal*f*-tansferases in *F. oxysporum*.**

| A. fumigatus | F. oxysporum | identity |
|:---:|:---:|:---:|
| gfsA | - | - |
| gfsB | EGU86692 | 35% |
| gfsC | EGU81169 | 25% |
| gfsD | EGU82964 | 28% |
| gfsE | EGU80311 | 30% |
| gfsF | EGU88219 | 24% |
| gfsG | EGU76406 | 27% |
| gfsH | EGU82964 | 36% |

BLASTp searches of the *A. fumigatus* galactofuranosyltransferases A-H in the genome of *F. oxysporum* 5176 (https://blast.ncbi.nlm.nih.gov). No GfsA homolog was found in the genome of EGU82964. EGU82964 encodes a homolog protein of GfsD (28% identity) and GfsH (36% identity).

with GfsA [6], were found in the genome of *F. oxysporum* 5176, the *gfsB* gene (EGU86692) was chosen to be further investigated in this study. The *gfsB* gene encodes a predicted protein (494 amino acids) sharing 35% of identity with *A. fumigatus* GfsB (Table 1).

The *ugmA*, *ugmB* and *gfsB* single mutants were constructed as described in Materials and methods. Using hygromycin resistance as a selective marker, transformants of the Δ*ugmA*, Δ*ugmB* and Δ*gfsB* were screened. The complemented strains of Δ*ugmA* were obtained with G418 sulfate screening. At the same time, the double mutant Δ*ugmA*Δ*ugmB* was also constructed. The complemented strains Re*umgA* and Re*umgB* were obtained by re-introducing *ugmA* and *ugmB* into the double mutant. All mutant strains were verified by PCR and Southern blotting (S2 Fig).

## Phenotypes of the mutants

As compared with the WT or complemented strains grown on PDA, the Δ*ugmA* mutant showed a slightly reduced growth and the Δ*ugmB* did not exhibit any difference, while the double mutant exhibited a severely retarded growth (Fig 1A). In contrast, the Δ*gfsB* mutant did not show any growth phenotype on PDA or MM (Fig 1B). After cultivation on PDA at 28˚C for 7 days, the Δ*ugmA* mutant produced less conidia than the WT, while the double mutant produced only very few conidia (Fig 2A and 2B). The number of conidia produced by the Δ*gfsB* mutant was reduced by 28% (Fig 2C). The germination of the Δ*ugmA* and the double mutant was delayed for 2 h and 4 h, respectively (Fig 3A), while the germination of the Δ*gfsB* mutant was similar to that of the WT (Fig 3B). Theoretically, the phenotypes of the DM-Re*ugmB* strain should be similar with the Δ*ugmA* strain. However, in our study we found that complementation of the double mutant strain with the *ugmB* gene did not restore the growth and conidiation to the levels similar to the Δ*ugmA* strain (Figs 1A and 2), which might be due to a lower activity of the plasmid we used for complementation of the double mutant.

On PDA medium supplemented with calcofluor white or Congo red, both Δ*ugmA* and the double mutant showed increased sensitivities. The double mutant was more sensitive to calcofluor white and Congo red than the Δ*ugmA*, while the sensitivity of the Δ*ugmB* mutant was similar to that of the WT (Fig 4A). When the *ugm* mutants were incubated on MM medium, both Δ*ugmA* and double mutant were hypersensitive to calcofluor white and Congo red. The sensitivity of the Δ*ugmA* to calcofluor white and Congo red was able to be rescued by the addition of 1.2 M sorbitol, while the double mutant not. These results suggest that the cell wall

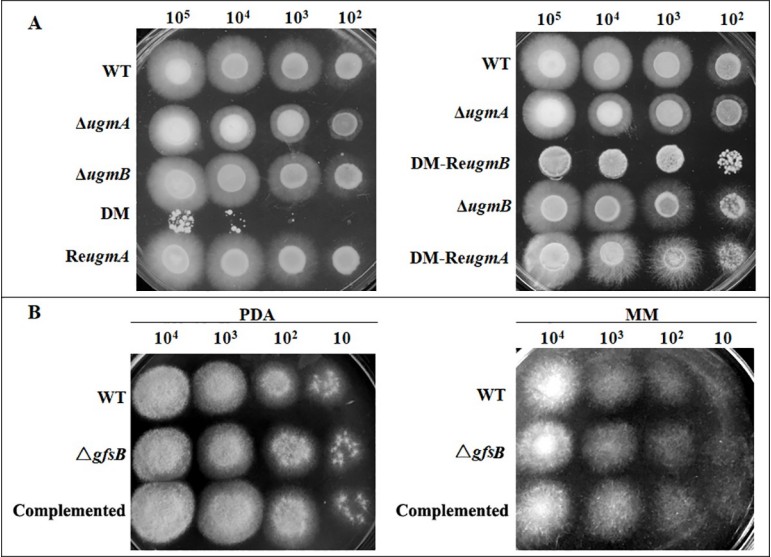

**Fig 1. Growth of the mutant strains.** Serial dilutions of conidia of $10^5$–$10^2$ from the wild-type (WT), mutants and complemented strains were grown on PDA (A) or MM (A and B) plates at 28°C for 7 days. DM, double-deletion mutant; DM-Re*ugmB*, double mutant with a copy of re-introduced *ugmB*; DM-Re*ugmA*, double mutant with a copy of re-introduced *ugmA*.

defect of the double mutant is more severe than that of the Δ*ugmA*. In contrast, the Δ*gfsB* mutant did not show any increased sensitivity to calcofluor white or Congo red (S3 Fig).

These results suggest that the *ugmA* contibutes to the cell wall integrity, hyphal growth and conidiation, whereas the *gfsB* is not required.

## Synthesis of Gal*f*-containing sugar chains in the mutants

As the Δ*ugmB* mutant did not showed any phenotypic changes and the Δ*ugmA* mutant exhibited defects in growth, cell wall integrity and conidiation as compared with the WT, it is likely

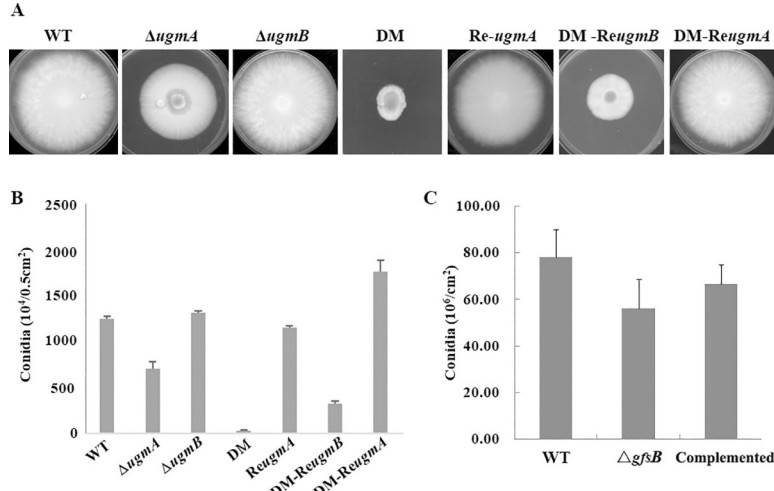

**Fig 2. Conidiation of the mutants.** $1×10^5$ of freshly conidia were inoculated at the central of plate, growth of the colonies were evaluated after 7-days culture at 28°C. The conidia produced by the strains were quantified in static cultures grown in continuous dark ($×10^4$/0.5 cm$^2$). The experiment was repeated for three times. Results are presented as mean ± SD (P ≤ 0.05). DM, double mutant; DM-Re*ugmB*, double mutant with a copy of re-introduced *ugmB*; DM-Re*ugmA*, double mutant with a copy of re-introduced *ugmA*.

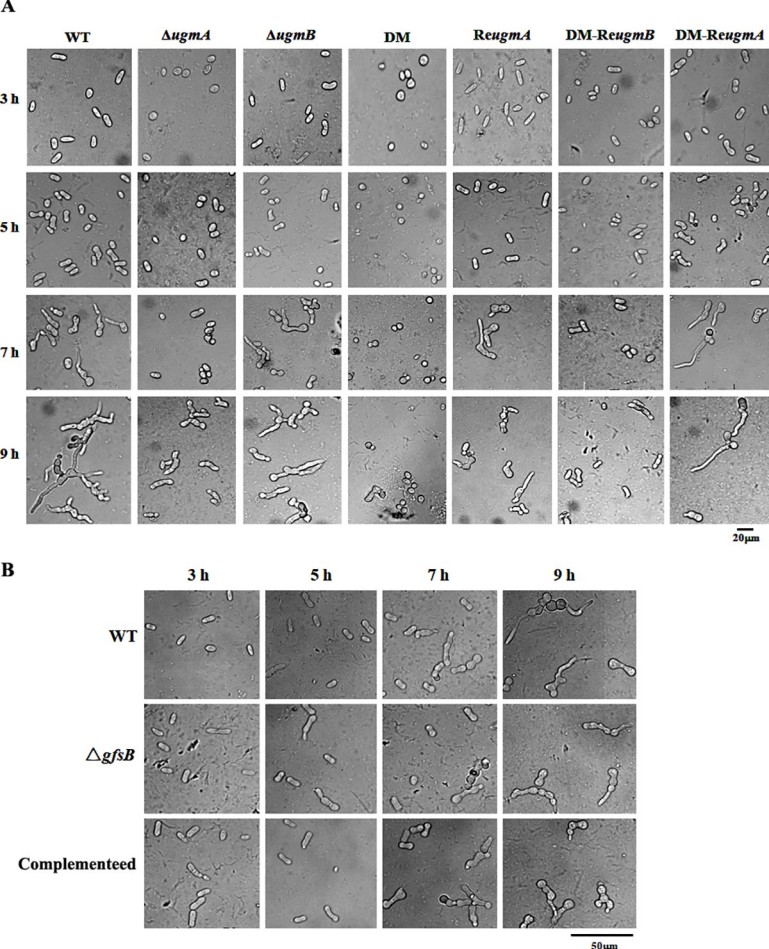

**Fig 3. Germination of the mutants.** $1\times10^5$/ml of freshly harvested conidia were inoculated onto the glass slide with double concave in 50 μl PDB liquid medium and incubated at 28°C, the slides were taken out at the indicated time points and observed with a microscope (400×). DM, double-deletion mutant. DM, double mutant; DM-Re*ugmB*, double mutant with a copy of re-introduced *ugmB*; DM-Re*ugmA*, double mutant with a copy of re-introduced *ugmA*.

that the *ugmA* can compensate for the loss of the *ugmB* gene. To testify this hypothesis, we analyzed the expression of the *ugmA* and *ugmB* genes in the WT and mutant strains. RT-PCR analysis revealed that the *ugmA* was expressed during hyphal growth (6–24 h) and highest expression occurred at log-phase (12 h), while the *ugmB* was expressed during conidiation (after 24 h) in the WT (Fig 5A), which suggests that UgmA contributes to the hyphal growth whereas UgmB contributes to the formation of conidiospores. In the Δ*ugmB* mutant the expression of the *ugmA* was increased during hyphal growth and conidiation with the highest expression level at 24 h (Fig 5B), while the expression of the *ugmB* was remarkably increased (0.2–10 folds) in the Δ*ugmA* mutant (Fig 5C). As the Δ*ugmB* mutant did not show any phenotype, it is reasonable to conclude that UgmA can completely compensate for the loss of UgmB, while UgmB could only partially complement the function of UgmA. Our results indicate that UgmA is more important for Gal*f*-containing sugar synthesis during vegetative growth of *F. oxysporum* f. sp. *cucumerinum*.

As shown in Fig 6, when the mutants were detected with the galactomannan-specific antibody AB135-8 that specifically recognizes galactofuranose [33], weak signals were detected

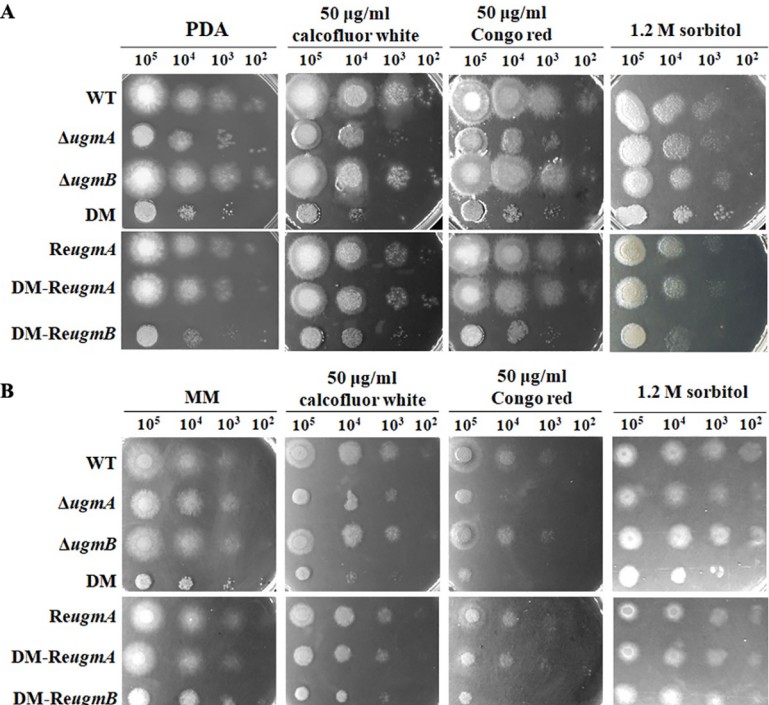

**Fig 4. Sensitivities of the mutants to cell wall stressors.** A serial dilution of conidia of $1 \times 10^5 - 1 \times 10^2$ of the WT, the mutants and complemented strains were inoculated on PDA (A) or MM (B) agar plates supplemented with 50 μg/ml Congo red or 50 μg/ml calcofluor white. Strains were cultured at 28˚C. DM, double mutant; DM-Re*ugmB*, double mutant with a copy of re-introduced *ugmB*; DM-Re*ugmA*, double mutant with a copy of re-introduced *ugmA*.

with the Δ*ugmA* and blurry signals were detected with the double mutant, while the Δ*gfsB* mutant was similar to the WT. The results reveal that both UgmA and UgmB are required for the synthesis of Gal*f*-containing sugars in *F. oxysporum* f. sp. *cucumerinum*, but GfsA does not.

## Virulence of the mutants

To evaluate the significance of the *ugmA*, *ugmB* and *gfsB* genes for the virulence of *F. oxysporum* f. sp. *cucumerinum*, $2 \times 10^5$ conidia/ml of each strain were inoculated to the cucumber seedling roots as described in Materials and methods. The infected seedlings were cultured at

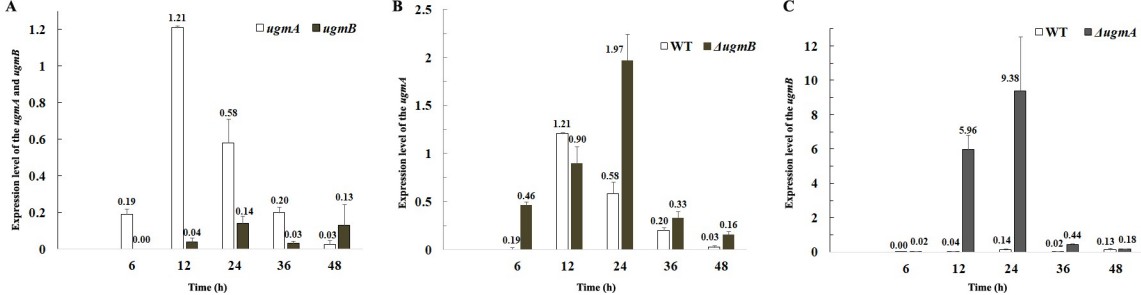

**Fig 5. Expression of the *ugm* genes in the WT and mutant strains.** Freshly harvested conidia were cultured in the PDB medium for 6, 12, 24, 36 and 48 hours. The hyphae were harvested and ground under liquid nitrogen. RT-PCR analysis was performed as described in Materials and methods. In A, expression level of the *ugmA* and *ugmB* in the WT strain; in B, expression level of the *ugmA* in the Δ*ugmB* mutant; and in C, expression level of the *ugmB* in the Δ*ugmA* mutant. All reactions were performed in triplicate. The relative mRNA expressions were calculated according to $2^{-\Delta\Delta Ct}$ with the transcription elongation factor gene (EF) as the reference.

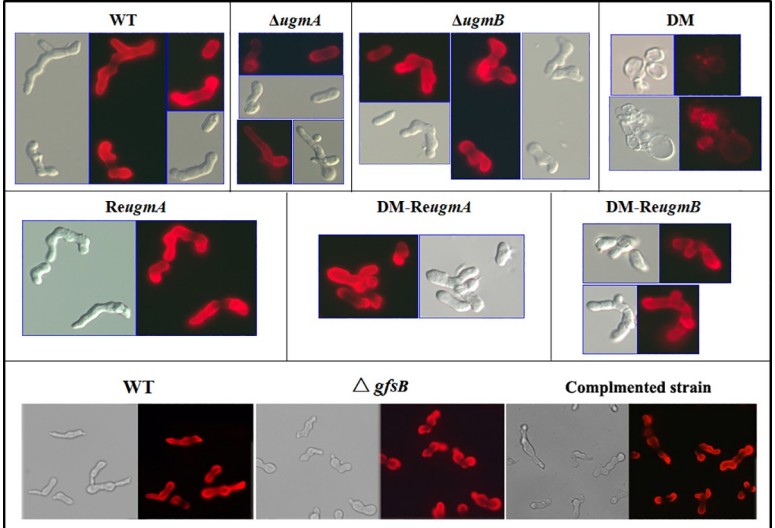

**Fig 6. Detection of the galactomannan in the mutants with antibody AB135-8.** Resting conidia were germinated for 8 h in PDB liquid medium and fixed with 2.5% pformaldehyde (PFA) overnight at 4˚C. After fixation, cells were washed with 0.2 M glycine in PBS for 5 min, then blocked with 5% BSA in PBS for 1 h. Cells were incubated with the anti-galactomannan monoclonal antibody at 1:10 dilution in 5% BSA in PBS for 1 h at room temperature. After washing with PBS-BSA 1%, cells were incubated with a goat Dylight 549-conjugated antibody directed against mouse IgG (H+L) diluted 1:2000 in BSA/PBS. After washing in PBS, cells were visualized with an inverted fluorescence light microscope. DM, double mutant; DM-R*eugmB*, double mutant with a copy of re-introduced *ugmB*; DM-R*eugmA*, double mutant with a copy of re-introduced *ugmA*.

28˚C in green house. Apparently, as shown in Fig 7A, the cucumber seedlings infected by the Δ*ugmA* and double-deletion mutant showed a slight wilt syndrome, indicating that the virulence of these two mutants was greatly attenuated. Calculation of the disease severity index (DSI) [35] also revealed that the DSI of both Δ*ugmA* and double mutant were 25% as compared with 70.8% for the WT and 18.7% for mock infected plants, while the virulence of the Δ*ugmB* mutant was similar to the WT (Fig 7E). These results indicated that the *ugmA* has a direct function in virulence of *F. oxysporum* f. sp. *cucumerinum*.

*F. oxysporum* f. sp. *cucumerinum* infects the plants through the roots. As shown in Fig 7B, the main roots from the plants infected by the WT strain and Δ*ugmB* strains showed brown color, a rotten appearance and fewer lateral roots, while the main roots infected by the Δ*ugmA* and double mutants suffered much less than those of the WT. Examination of root sections under a stereoscopic microscope (Fig 7C) and qPCR analysis (Fig 7D) revealed that the vascular bundles of roots that were infected with the Δ*ugmA* or the double mutant were comparable to mock infected roots, while those infected by the WT and the Δ*ugmB* strain were severely brown stained in their cortex and vascular bundles. These results demonstrate that *ugmA* was required for the virulence of *F. oxysporum* f. sp. *cucumerinum* in cucumber. Interestingly, the Δ*gfsB* mutant also exhibit a reduced virulence although no visible phenotypes were detected in the other assays (Fig 7A–7C and 7E).

## Discussion

*Fusarium* species are common plant pathogens [25], but some species are also pathogenic for humans, particularly *F. solani*, *F. oxysporum*, and *F. moniliforme*, causing superficial infections in immnocompetent individuals, and severe and disseminated disease in

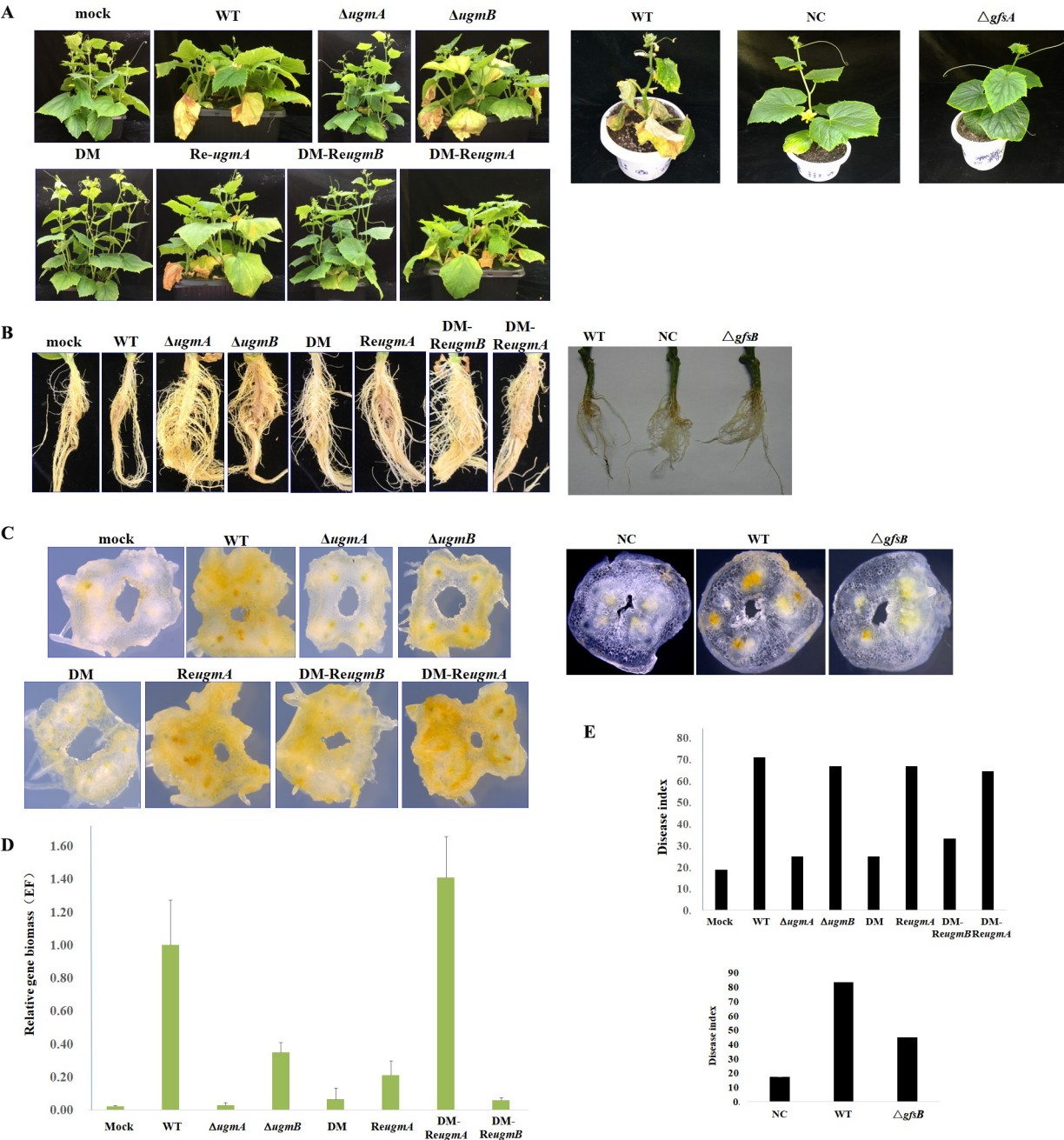

**Fig 7. Virulence of the mutant strains.** In A and B, the germinated cucumber seeds in plastic pots were grown in the greenhouse at approximately 28˚C for a 16 h photoperiod until the two-true leaves stage. The infected seedlings were analyzed after 45 days. Twelve plants were used for each treatment. In C, the root cross sections of infected plants were observed with M205A stereoscopic microscope (Leica Microsystems) (×20). In D, to quantify the relative mycelial biomass of the fungal strains in cucumber roots, DNAs were extracted from the cucumber roots cultured for 30 days after infection and subjected to quantitative PCR (qPCR) using primers EF1a-1/-2 for the transcription elongation factor gene of *F. oxysporum* as described in Materials and methods. In E, disease severity index (DSI) was calculated as previously described from 2 weeks of inoculation up to 45 days. The plant infection experiments were repeated for three times.

immunocompromised hosts [36,37]. *F. oxysporum* Gal*f*-containing sugar chain consists of a main chain containing β1,6-linked Gal*f* residues with side chains containing α1,2-linked Glc*p*, β1,2-linked Man*p* and β-Man*p* terminal nonreducing end units, which is different

from that of *A. fumigatus* and important for the activation of macrophage mechanisms [26]. As Gal*f* is not found in mammalian and plant cells [12], inhibition of the biosynthetic pathway of Gal*f*-containing sugar chains is an ideal target for the development of medicines and agricultural chemicals, which can specifically inhibit *F. oxysporum* without any side effects.

The genes encoding the UDP-Gal*p* mutase have recently been identified in several fungal species such as *Cryptococcus neoformans*, *A. fumigatus*, *A. nidulans* and *A. niger* [10,14,21]. In *A. fumigatus*, abrogation of Gal*f*-synthesis causes attenuated virulence and reduced growth [16,19,20,23], indicating that a lack of Gal*f*-containing sugar chains affects normal hyphal growth, cell wall structure and conidiation in filamentous fungi.

In this report, we used *F. oxysporum* f.sp. *cucumerinum* as a model species to evaluate the biologial functions of Gal*f*-containing sugar chain in *F. oxysporum*. Two UGM genes (*ugmA* and *ugmB*) were identified in the genome of *F. oxysporum*. Phentypes of the single and double mutant of the UGM genes were investigated. Based our results, UgmA was required for vegetative growth and UgmB contributed to conidiation in *F. oxysporum* f.sp. *cucumerinum*. Although the *ugmB* was highly expressed in the Δ*ugmA*, only a weak signal was detected by antibody AB135-8 on the cell wall of the Δ*ugmA* mutant (Figs 5C and 6), suggesting that the residual Gal*f*-containing sugars were not enough for maintaining the mycelial cell wall integrity. On the other hand, the elevated expression of the *ugmA* could compensate for the loss of the *ugmB* (Fig 5B). These results suggest that UgmA is able to complement the function of UgmB during conidiation, whereas UgmB is not able to function as UgmA during vegetaive growth stage. Although the mechanism remains unclear, it is likely that UgmA and UgmB differ in their activities. Further investigation should be carried out to characterize these two enzymes.

We found that both Δ*ugmA* and double mutant showed an attenuated virulence, suggesting UGMs, especially UgmA, play an important role in virulence. Therefore, UgmA represents an attractive target for the development of new drugs.

In *A. fumigatus*, the cell wall galactomannan (GM) is composed of a main α1,2/α1,6-mannan chain and a number of side chains of β1,5/β1,6-linked Gal*f* residues in its cell wall [3]. Eight putative *gfs* genes are identified in the genome of *A. fumigatus*, including *gfsA* (Afu6g02120), *gfsB* (Afu4g13710), *gfsC* (Afu4g10170), *gfsD* (Afu6g00520), *gfsE* (Afu3g07220), *gfsF* (Afu3g00170), *gfsG* (Afu2g17320) and *gfsH* (Afu8g07260). All eight Gfs proteins can be phylogenetically classified into two groups: GT31-A and GT31-B [6]. Recently, GfsA is biochemically characterized as the enzyme responsible for the transfer of β1,5-linked Gal*f* to the side chain of GM [20]. As GfsA, GfsB and GfsC are belonged to GT31-A family, GfsB and GfsC are also predicted as β1,5-galactofuranosyltransferases similar to GfsA [20]. GfsA, GfsB and GfsC are also identified in the genome of *A. niger* and *A. nidulans* [19,24]. In contrast to *A. fumigatus*, *A. niger* and *A. nidulans*, no GfsA homolog was found in the genome of *F. oxysporum* (Fig 8), suggesting that the linkage of cell wall Gal*f* residues are different in *F. oxysporum*. Indeed, it has been reported that *Fusarium* produces cell wall β1-6-linked Gal*f* residues [26]. Therefore, it is not surprised that we were unable to find GfsA homolog in *F. oxysporum*. We only found two GT31-A family members in *F. oxysporum*, GfsB (EGU86692) and GfsC (EGU81169), which share 35% and 25% of identity with *A. fumigatus* GfsB and GfsC, respectively (Table 1 and Fig 8). In this study, the significance of the *gfsB* gene was evaluated. Deletion of the *gfsB* gene did not show any visible phenotype, however, the Δ*gfsB* mutant displayed a reduced virulence. Although its substrate specificity needs to be characterized, it is no doubt that GfsB is also an ideal drug target for the development of novel drugs against *F. oxysporum*.

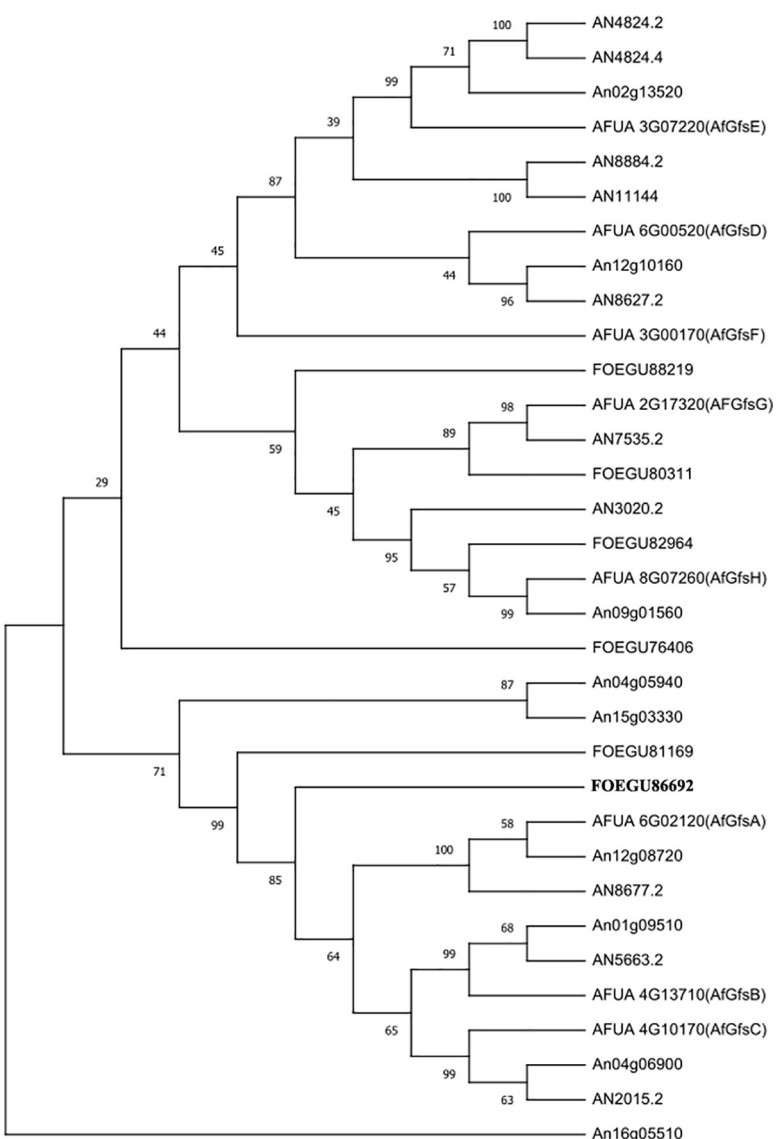

**Fig 8. Phylogenetic tree of putative galactofuranosyltransferase from *A. nidulans* and *A. fumigatus*.** Protein sequences were retrieved from AspGD (https://www.ncbi.nlm.nih.gov/genome/?term=Aspergillus+fumigatus) and DNAman2.0 was used to make the homology tree. % of homology between the proteins is indicated.

## Supporting information

**S1 Fig. Alignment of UDP-galactopyranose mutases.** BLASTp searches of the *A. fumigatus* UDP-galactopyranose mutase GlfA/Ugm1 (AFU_3G12690) in the genome of *F. oxysporum* 5176 identified three homologous genes, *ugmA* (EGU88002.1) and *ugmB* (EGU83790). The *ugmA* encodes a predicted protein (519 amino acids) sharing 80% of identity with *A. fumigatus* GlfA, while the *ugmB* encodes a protein (507 amino acids) sharing 71% of identity with *A. fumigatus* GlfA.
(TIF)

**S2 Fig. Souhtern blotting of the single and double mutant of the *ugmA* and *ugmB* genes.** Southern blotting was carried by detecting *Hind*III-digested fragments with a probe in the

upstream non-coding region of the *ugmA* or *ugmB* gene.
(TIF)

**S3 Fig. Sensitivities of the mutant to cell wall stressors.** A serial dilution of conidia of $1\times10^4$–$1\times10^1$ of the WT, the mutant and complemented strain were inoculated on PDA agar plates supplemented with 40 μg/ml calcofluor white or 40 μg/ml Congo red. Strains were cultured at 28˚C.
(TIF)

**S1 Table. Oligonucleotides used in this study.**
(PDF)

## Acknowledgments

We are grateful to Prof. Huishan Guo and Prof. Zhaosheng Kong, Institute of Microbiology, Chinese Academy of Sciences, for their help in virulence test.

## Author Contributions

**Conceptualization:** Cheng Jin.

**Data curation:** Hui Zhou, Yueqiang Xu.

**Formal analysis:** Hui Zhou, Yueqiang Xu.

**Funding acquisition:** Cheng Jin.

**Investigation:** Hui Zhou, Yueqiang Xu.

**Methodology:** Hui Zhou, Yueqiang Xu, Frank Ebel, Cheng Jin.

**Resources:** Hui Zhou, Yueqiang Xu, Frank Ebel.

**Software:** Hui Zhou.

**Validation:** Hui Zhou.

**Visualization:** Hui Zhou, Cheng Jin.

**Writing – original draft:** Cheng Jin.

**Writing – review & editing:** Frank Ebel, Cheng Jin.

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
