## [Decision Letter · Decision Letter 0]

24 Feb 2021

PONE-D-21-02191

Galactofuranose (Galf)-containing sugar chain is essential for the growth, conidiation and virulence of F. oxysporum f.sp. cucumerinum

PLOS ONE

Dear Dr. Jin,

Thank you for submitting your manuscript to PLOS ONE. After careful consideration, we feel that it has merit but does not fully meet PLOS ONE’s publication criteria as it currently stands. Therefore, we invite you to submit a revised version of the manuscript that addresses the points raised during the review process.

We look forward to receiving your revised manuscript.

Kind regards,

Zonghua Wang, Ph.D.

Academic Editor

PLOS ONE

Journal Requirements:

3. Please include a copy of Table 1 which you refer to in your text on page 18.

4.We noticed you have some minor occurrence of overlapping text with the following previous publication(s), which needs to be addressed:

https://www.tandfonline.com/doi/full/10.1080/09168451.2017.1422383

https://onlinelibrary.wiley.com/doi/full/10.1111/j.1462-5822.2009.01352.x

In your revision ensure you cite all your sources (including your own works), and quote or rephrase any duplicated text outside the methods section. Further consideration is dependent on these concerns being addressed.

Reviewers' comments:

Reviewer's Responses to Questions

**Comments to the Author**

1. Is the manuscript technically sound, and do the data support the conclusions?

Reviewer #1: Yes

Reviewer #2: Yes

2. Has the statistical analysis been performed appropriately and rigorously? 

Reviewer #1: No

Reviewer #2: Yes

3. Have the authors made all data underlying the findings in their manuscript fully available?

Reviewer #1: Yes

Reviewer #2: Yes

4. Is the manuscript presented in an intelligible fashion and written in standard English?

Reviewer #1: Yes

Reviewer #2: Yes

5. Review Comments to the Author

Reviewer #1: Dear Edtor,

The paper by Zhou et al. describes the role of cell wall related galactofuranose containing sugar chain in Fusarium oxysporum. To address the role of Galf residues, the author have made targeted deletion of two UDP-galactopyranose mutases (ugmA and ugmB) and a single galactofuranose transferase (gfsA). The analysis of gfsA is somewhat premature especially since in other studies the redundancy of gfs genes has been shown.

The main contribution of the paper is that it shows the impact on no galf on the virulence but the weak point is that the growth of the double mutant is severely affected and reduced which questions whether the lack of virulence can be directly related to the absence of galF or this reduced growth. This has not been addressed sufficiently in the paper.

In the title and throughout the paper the author state that Galf-containing sugar chain are essential for growth….(titel) or essential for cell wall integrity (line 31) etc. Clearly, Galf is not essential for growth (de mutant lacking Galf is viable) so please do not use the word essential (throughout the manuscript). Use for example … Galf biosynthesis contributes ….

Line 28/29 Rephrase. …. suggesting that UgmA and UgmB are redundant and that they can both contribute to synthesis of UDP-Galf.

Line 72 and further. Include and cite also the work in Aspergillus niger.

It is not clear if the Fusarium gfsA gene is most homologous to the A fumigatus gfsA gene and why this gfs gene was choosen for further analysis. E.g. Expression analysis (as being done for ugmA and ugmB) might be a reason and it is not clear why the expression of gfsA has not been taken along. Include also a phylogenic analysis of the six gfs genes. Are they all grouping together and are likely to encode Galf transferases?

The term revertant for complemented strains is misleading. Please use the word complemented strain. Revertant suggest a spontaneous mutation occurring that reverts the phenotype. Also adjust in Figures.

Line 130 please include some statistical analysis on the data. “the number of conidia was slightly reduced” can not be included. Give a number here and indicate if this is relevant (Student TTest or equivalent)

Fig 5. Check labeling and figure legend. Expression level (fold) is not clear. Label figure A, B and C and explain clearly what we see.

Is it true that the strain start to sporulate in liquid cultures after 24 hours of growth? The authors should explain why we see still a red signal in de double mutant. In other species is has been shown that deletion of ugmA and B abolish galf biosynthesis.

Line 286: A. fumigatus also contains three gfs genes (see ne.g. Arentshorst et al., 2019). Adjust the discussion and include discussion related to the phylogenetic analysis.

Reviewer #2: Comments:

In this work，Zhou et al. investigated three uncharacterized genes from a plant pathogenic fungus, F. oxysporum f.sp. cucumerinum, which putatively encode enzymes involved in the biosynthesis of Galactofuranose (Galf). Two of these enzymes are the homologs of Aspergillus fumigatus UgmA named as UgmA and UgmB, while the third one shears the sequence similarity with Aspergillus fumigatus GfsA named as GfsA. Authors found that deletion of ugm genes, especially the ugmA, led to a reduced production of Galf-containing sugar chains, resulting growth defect and impaired conidiation phenotypes in F. oxysporum f.sp. cucumerinum. Most importantly, the ugmA deletion mutant as well as the ΔgfsA strain lost their pathogenicity in cucumber plantlets. Thus, authors claimed that these two genes can be the targets for the development of new anti-Fusarium agents. Results and the conclusion seem reasonable, and show certain importance on the research filed of pathogenic fungi. However, there are several mistakes and inappropriate presentations in the manuscript, especially in the result part. These may simply be considered as careless miss, but this reviewer believes that some of them will critically influence the conclusion of this work. If the manuscript would be published in the journal, revision is necessary in regards to the following points.

Major points:

1) In Figure 4:

As described in the figure legend, the DM-ReugmA represents the double mutant (ΔugmAΔugmB) with a copy of re-introduced ugmA; and DM-ReugmB is the double mutant with a copy of re-introduced ugmB. If these are true, then the DM-ReugmA strain should has the similar growth phenotype with the ΔugmB strain, and the DM-ReugmB stain should grow similar with the ΔugmA strain. However, in the Figure 4A and 4B, DM-ReugmB strain clearly show a better growth phenotype than the DM-ReugmA, while the growth of the ΔugmA strain is more sensitive to calcofluor white and Congo red than ΔugmB strain. How can explain these different phenotypes?

2) In Figure 5:

This figure is very hard to follow. Three panels have been presented here. As reviewer’s understanding, the upper panel of the left side represents the expression levels of ugmA and ugmB genes in the wild type (WT) strain. Here, the empty bar indicates the expression level of ugmA gene and the solid bar shows that of ugmB. This panel is clear and easy to understand because the Y-axis is labelled as “expression level”. On the other hand, the lower panel should represent the expression level of ugmA gene in both WT and ΔugmB strains. In this panel, the empty bar indicates the WT strain and the solid bar represents the ΔugmB strain. However, its Y-axis still be labelled as “expression level”. It is confusing to the readers that same empty or solid bar indicates the different things in a same figure. These two panels should be separated into Figure 5A and 5B, and each of it should be given a different subtitle to describe their difference. Same with the panel of right side, it should be presented as Figure 5C, and has its own subtitle indicating the ugmB expression in WT and ΔugmA strain.

3) In Figure 7D:

Same with in Figure 4, The EFs of DM-ReugmA and DM-ReugmB strains are not consistent with those of ΔugmB and ΔugmA strains.

Minor points:

1) In the introduction: Line 75-76 of page 4 describes “In A. nidulans, deletion of the ugeA, the gene encoding UGM, leads to highly branched hyphae and reduced conidiation”. The ugeA should be the gene encoding a UGE (UDP-glucose 4-epimerase), not the UGM (UDP-galactopyranose mutase)?

2) In Figure 1A and 2: Different levels of growth and conidiation can be clearly observed between the DM-ReugmB and ΔugmA strains. These results should be indicated, and the biological reasons should be discussed.

3) The galactomannan-specific antibody AB135-8 was used in Figure 6 to staining the cell wall. Dose this antibody specifically recognize the Galf-containing structure? It should be indicated or emphasized in the manuscript.

6. PLOS authors have the option to publish the peer review history of their article (what does this mean?). If published, this will include your full peer review and any attached files.

Reviewer #1: **Yes: **Arthur F.J. Ram

Reviewer #2: No

---

## [Author Response · Author response to Decision Letter 0]

28 Feb 2021

Response to Reviewers' comments:

Reviewer #1: Dear Edtor,

The paper by Zhou et al. describes the role of cell wall related galactofuranose containing sugar chain in Fusarium oxysporum. To address the role of Galf residues, the author have made targeted deletion of two UDP-galactopyranose mutases (ugmA and ugmB) and a single galactofuranose transferase (gfsA). The analysis of gfsA is somewhat premature especially since in other studies the redundancy of gfs genes has been shown.

The main contribution of the paper is that it shows the impact on no galf on the virulence but the weak point is that the growth of the double mutant is severely affected and reduced which questions whether the lack of virulence can be directly related to the absence of galF or this reduced growth. This has not been addressed sufficiently in the paper.

In the title and throughout the paper the author state that Galf-containing sugar chain are essential for growth….(titel) or essential for cell wall integrity (line 31) etc. Clearly, Galf is not essential for growth (de mutant lacking Galf is viable) so please do not use the word essential (throughout the manuscript). Use for example … Galf biosynthesis contributes ….

It’s good suggestion. We have changed the title and the text by taking the reviewer’s suggestion. 

Line 28/29 Rephrase. …. suggesting that UgmA and UgmB are redundant and that they can both contribute to synthesis of UDP-Galf.

It has been rephrased.

Line 72 and further. Include and cite also the work in Aspergillus niger.

Yes, we have added A. niger work.

It is not clear if the Fusarium gfsA gene is most homologous to the A fumigatus gfsA gene and why this gfs gene was choosen for further analysis. E.g. Expression analysis (as being done for ugmA and ugmB) might be a reason and it is not clear why the expression of gfsA has not been taken along. Include also a phylogenic analysis of the six gfs genes. Are they all grouping together and are likely to encode Galf transferases?

This is a good comment. Indeed, we initiated this work about 5-6 years ago. At that time, we serached the genome sequence with gfs, only one gene was identified. So we named this gene as gfsA. Later we found 6 gfs genes in F. oxysporum, but we did not change the name of gfsA. This reviewer gave a good suggestion. So we went back to check the phylogenic tree. It turns out that it would more accurate to name this gene as gfsB. To clear describe this, we added Table 1 and Fig 8 in our manuscript. Also, we added the description in the results and discussion. Please check them out.

The term revertant for complemented strains is misleading. Please use the word complemented strain. Revertant suggest a spontaneous mutation occurring that reverts the phenotype. Also adjust in Figures.

All revertant used in this manuscript has been changed into complemented strain.

Line 130 please include some statistical analysis on the data. “the number of conidia was slightly reduced” can not be included. Give a number here and indicate if this is relevant (Student TTest or equivalent)

It has been added.

Fig 5. Check labeling and figure legend. Expression level (fold) is not clear. Label figure A, B and C and explain clearly what we see.

We have re-made this figure and re-written the figure legend.

Is it true that the strain start to sporulate in liquid cultures after 24 hours of growth? The authors should explain why we see still a red signal in de double mutant. In other species is has been shown that deletion of ugmA and B abolish galf biosynthesis.

Yes, regularly the strain will reach conidiation stage after 24 h under our experiment condition. The red signal still can be obserbved in double mutant, but this signal is weak and not continuously distributed on cell wall, which might be due to non-specific binding of antibody.

Line 286: A. fumigatus also contains three gfs genes (see ne.g. Arentshorst et al., 2019). Adjust the discussion and include discussion related to the phylogenetic analysis.

Yes, we have adjusted the discussion.

Reviewer #2: Comments:

In this work，Zhou et al. investigated three uncharacterized genes from a plant pathogenic fungus, F. oxysporum f.sp. cucumerinum, which putatively encode enzymes involved in the biosynthesis of Galactofuranose (Galf). Two of these enzymes are the homologs of Aspergillus fumigatus UgmA named as UgmA and UgmB, while the third one shears the sequence similarity with Aspergillus fumigatus GfsA named as GfsA. Authors found that deletion of ugm genes, especially the ugmA, led to a reduced production of Galf-containing sugar chains, resulting growth defect and impaired conidiation phenotypes in F. oxysporum f.sp. cucumerinum. Most importantly, the ugmA deletion mutant as well as the ΔgfsA strain lost their pathogenicity in cucumber plantlets. Thus, authors claimed that these two genes can be the targets for the development of new anti-Fusarium agents. Results and the conclusion seem reasonable, and show certain importance on the research filed of pathogenic fungi. However, there are several mistakes and inappropriate presentations in the manuscript, especially in the result part. These may simply be considered as careless miss, but this reviewer believes that some of them will critically influence the conclusion of this work. If the manuscript would be published in the journal, revision is necessary in regards to the following points.

Major points:

1) In Figure 4:

As described in the figure legend, the DM-ReugmA represents the double mutant (ΔugmAΔugmB) with a copy of re-introduced ugmA; and DM-ReugmB is the double mutant with a copy of re-introduced ugmB. If these are true, then the DM-ReugmA strain should has the similar growth phenotype with the ΔugmB strain, and the DM-ReugmB stain should grow similar with the ΔugmA strain. However, in the Figure 4A and 4B, DM-ReugmB strain clearly show a better growth phenotype than the DM-ReugmA, while the growth of the ΔugmA strain is more sensitive to calcofluor white and Congo red than ΔugmB strain. How can explain these different phenotypes?

Sorry, we have mis-labeled the DM-ReugmB and DM-ReugmA. Now it has been revised in Fig 4.

2) In Figure 5:

This figure is very hard to follow. Three panels have been presented here. As reviewer’s understanding, the upper panel of the left side represents the expression levels of ugmA and ugmB genes in the wild type (WT) strain. Here, the empty bar indicates the expression level of ugmA gene and the solid bar shows that of ugmB. This panel is clear and easy to understand because the Y-axis is labelled as “expression level”. On the other hand, the lower panel should represent the expression level of ugmA gene in both WT and ΔugmB strains. In this panel, the empty bar indicates the WT strain and the solid bar represents the ΔugmB strain. However, its Y-axis still be labelled as “expression level”. It is confusing to the readers that same empty or solid bar indicates the different things in a same figure. These two panels should be separated into Figure 5A and 5B, and each of it should be given a different subtitle to describe their difference. Same with the panel of right side, it should be presented as Figure 5C, and has its own subtitle indicating the ugmB expression in WT and ΔugmA strain.

We have made a new figure and re-written the legend. We hope this new figure will be eay to understand.

3) In Figure 7D:

Same with in Figure 4, The EFs of DM-ReugmA and DM-ReugmB strains are not consistent with those of ΔugmB and ΔugmA strains.

Yes, this was also mis-labeled. We have revised.

Minor points:

1) In the introduction: Line 75-76 of page 4 describes “In A. nidulans, deletion of the ugeA, the gene encoding UGM, leads to highly branched hyphae and reduced conidiation”. The ugeA should be the gene encoding a UGE (UDP-glucose 4-epimerase), not the UGM (UDP-galactopyranose mutase)?

It has been revised.

2) In Figure 1A and 2: Different levels of growth and conidiation can be clearly observed between the DM-ReugmB and ΔugmA strains. These results should be indicated, and the biological reasons should be discussed.

Well, it happens sometimes since we used different plasmids and selective marker to generate the complemented strain as descibed in Material and method. We have added a short discussion in the results.

3) The galactomannan-specific antibody AB135-8 was used in Figure 6 to staining the cell wall. Dose this antibody specifically recognize the Galf-containing structure? It should be indicated or emphasized in the manuscript.

This antibody specifically binds to Galf alone. We have changed the description in the text.

---

## [Decision Letter · Decision Letter 1]

31 Mar 2021

Galactofuranose (Galf)-containing sugar chain contributes to the hyphal growth, conidiation and virulence of F. oxysporum f.sp. cucumerinum

PONE-D-21-02191R1

Dear Dr. Jin,

We’re pleased to inform you that your manuscript has been judged scientifically suitable for publication and will be formally accepted for publication once it meets all outstanding technical requirements.

Kind regards,

Zonghua Wang, Ph.D.

Academic Editor

PLOS ONE

Additional Editor Comments (optional):

Reviewers' comments:

Reviewer's Responses to Questions

**Comments to the Author**

1. If the authors have adequately addressed your comments raised in a previous round of review and you feel that this manuscript is now acceptable for publication, you may indicate that here to bypass the “Comments to the Author” section, enter your conflict of interest statement in the “Confidential to Editor” section, and submit your "Accept" recommendation.

Reviewer #2: All comments have been addressed

2. Is the manuscript technically sound, and do the data support the conclusions?

Reviewer #2: Yes

3. Has the statistical analysis been performed appropriately and rigorously? 

Reviewer #2: N/A

4. Have the authors made all data underlying the findings in their manuscript fully available?

Reviewer #2: (No Response)

5. Is the manuscript presented in an intelligible fashion and written in standard English?

Reviewer #2: Yes

6. Review Comments to the Author

Reviewer #2: Authors have answered all my questions, and made corrections in the revised version. This reviewer has no further request,

7. PLOS authors have the option to publish the peer review history of their article (what does this mean?). If published, this will include your full peer review and any attached files.

Reviewer #2: **Yes: **GAO, XIAO-DONG

---

## [Editor Report · Acceptance letter]

16 Jul 2021

PONE-D-21-02191R1 

Galactofuranose (Gal*f*)-containing sugar chain contributes to the hyphal growth, conidiation and virulence of *F. oxysporum* f.sp. *cucumerinum*

Dear Dr. Jin:

I'm pleased to inform you that your manuscript has been deemed suitable for publication in PLOS ONE. Congratulations! Your manuscript is now with our production department. 

Kind regards, 

on behalf of

Prof. Zonghua Wang 

Academic Editor

PLOS ONE